


# Measurement report: An assessment of the impact of a nationwide lockdown on air pollution – a remote sensing perspective over India

Mahesh Pathakoti,[1]*Aarathi Muppalla,[2]Sayan Hazra,[3]Mahalakshmi D. Venkata,[1]Kanchana A. Lakshmi,[1]Vijay K. Sagar,[1]Raja Shekhar,[2]Srinivasulu Jella,[1]Sesha Sai M.V. Rama,[1] Uma Vijayasundaram[3]

[1]Analytics and Modelling Division, Land and Atmospheric Physics Division,Earth and Climate Systems Study Division,Atmospheric Chemistry Division, Earth and Climate Sciences Area,National Remote Sensing Centre (NRSC), Indian Space Research Organization (ISRO), Hyderabad-500037, India.
[2]Bhuvan Project Management and Software Evaluation Division, Bhuvan Geoportal and Data Dissemination Area, National Remote Sensing Centre (NRSC), Indian Space Research Organization (ISRO), Hyderabad-500037, India.
[3]Department of Computer Science, School of Engineering & Technology, Pondicherry University, ChinnaKalapet, Kalapet, Puducherry-605014, India

*Correspondence to*: Mahesh P (mahi952@gmail.com);

**Abstract.** The nationwide lockdown was imposed over India from 25[th] March to 31[st] May2020 with varied relaxations from phase-I to phase-IVto contain the spread of COVID-19. Thus emissions from industrial and transport sectorswere halted during lockdown (LD)which resulted in a significant reduction of anthropogenic pollutants. The first twolockdown phases werestrictly followed(phase-I and phase-II) and hence are considered as total lockdown (TLD) in this study. Satellite-based tropospheric columnar nitrogen dioxide (TCN) from the years 2015 to 2020, tropospheric columnar carbon monoxide (TCC) during 2019-2020 and aerosol optical depth ($AOD_{550}$) from the years 2014 to 2020 during phase-I and phase-II LDand pre-LD periods were investigated with observations from Aura/OMI, Sentinel-5P/TROPOMI, and Aqua-Terra/ MODIS satellite sensors.To quantify lockdown induced changesin TCN, TCC, and $AOD_{550}$, detailed statistical analysis was performed on de-trended data using student's paired statistical t-test.Results indicate that mean TCN levels over India showed a dip of 18% compared to the previous year and also against the 5-year mean TCN levelsduring the phase-I lockdown, which was found statistically significant (p-value <0.05) against the respective period. Furthermore, drastic changes in TCN levels were observed over hotspots namely the eastern region and urban cities. For example,there was a sharp decrease of 62%and 54% in TCN levels as compared to 2019 and against 5-years mean TCN levels over New Delhi with a p-value of 0.0002 (which is statistically significant) during total LD.The TCC levels werehigh in the North East (NE) region during the phase-I LD period, which is mainly attributed to the active fire counts in this region. However, lower TCC levels are observed in the same region due to the diminished fire counts during phase-II. Further, $AOD_{550}$is reduced over the country by ~16 %(Aqua





and Terra) from the 6-years (2014-2019) mean $AOD_{550}$levels, with a significantreduction (Aqua/MODIS 28%)observed overthe Indo-Gangetic plains (IGP) region with a p-value of <<0.05.However, an increase in $AOD_{550}$ levels (25% for Terra/MODIS, 15% for Aqua/ MODIS) was also observed over Central Indiaduring LD compared to the preceding year and found significant with a p-value of 0.03.This study also reports the rate of change of TCNlevels and $AOD_{550}$ along with statistical metrics during the LD period.

**Keywords:** COVID-19, lockdown, Satellite, TCN, TCC, $AOD_{550}$

## 1 Introduction

Following the outbreak of the novel coronavirus disease (COVID-19) and its declaration as pandemic by the World Health Organization (WHO) on 11[th] March 2020, several countries across the globe imposed national lockdowns to contain the pandemic (Tian et al., 2020). India confirmed its first COVID-19 case on 30[th] January 2020 with an exponential increase to 360 cases by 22[nd] March 2020 (https://www.mohfw.gov.in/). In an attempt to restrict this pandemic, the Indian government called for a 'Janata Curfew' on 22[nd] March 2020, followed by nationwide lockdown (LD) in phasedmanner starting from 25[th] March – 14[th] April 2020 (21days) as phase-I, 15[th] April – 3[rd] May 2020 (19 days) as phase-II, 4[th] May–17[th] May 2020(14 days) as phase-III and 18[th] May – 31[st] May in 2020(14 days)as phase-IV. Under this lockdown, 1.30 billion citizens of India were advised to stay in-doors,all the domestic and international flights, transport and industrial production were suspended, and only essential services were permitted.However, agriculture farming and its related sectors are permitted during phase-II as India is agricultural-dependent country. As indicated above, indoor emissions (cooking) and emissions from the emergency services still present in phase-I.For phase-II, crop residue burningis added in addition to phase-I emissions. Except these, restof the anthropogenic emissions from above sectors are completely shut during phase-I and phase-II. Thus, economic activities were greatly affected and hence there was a shortfall in net energy consumption by about 30% (https://www.ppac.gov.in/) during strict lockdown period (first two lockdown phases).

Air pollution has arisen as an environmental issue which is harmful to human health (Xu et al., 2020)and extends from local to global scale (Fang et al., 2009).The oxides of nitrogen (NO, $NO_2$) play important role in tropospheric chemistry and climate change. Exposure to $NO_2$ has been correlated with an increased rate of morbidity and subsequently increased rate of mortality (WHO, 2013). Global emissions of $NO_x$ (NO, $NO_2$) are primarily due to anthropogenic activities such as transportation (32% in India), industrial



activities (21% in India), thermal power plants (28% in India), biomass burning (19% in India)whereas the natural sources of $NO_x$ are soils and lightning (Biswal et al., 2020b). Thus, hotspots region of $NO_2$are thermal power plants, urban cities, and industrial regions. In addition to $NO_2$, carbon monoxide (CO) is also an important trace gas in the troposphere and is the main precursor of secondary pollutant ozone in $NO_x$ rich environment. Though CO is not a direct greenhouse gas, it has a global warming potential because of its effects on the lifetime of several greenhouse gases. The natural and anthropogenic sources of CO are forest fire,biofuel burning, volcanic activities, and incomplete combustion of fossil fuels, oil, coal, woods, natural gas, and oxidation of hydrocarbons. However, significant amount of contribution to CO is from the anthropogenic emissionsonly (Beig et al., 2020). Harmful effects of CO are dizziness, headaches, stomach-ache, confusion, tiredness.CO is tracer of air pollution due to its lifetimeof about ~1-2 months(Filonchyk et al., 2020).

Natural and anthropogenic activities are responsible for aerosols in the atmosphere. Anthropogenic activities over South Asia have caused considerable changes in aerosol composition and loading. Fine mode aerosols ($PM_{2.5}$) are mainly from gas to particle conversions which are from biogenic and anthropogenic emissions. Coarse mode aerosols (particles with diameter larger than 10 μm) arise from natural sources namely deserts, oceans, volcanoes and biosphere with less contribution from anthropogenic activities. Over the ocean surface, the natural global aerosol mass is controlled by sulphate, sea salt, and dustaerosols (David et al., 2018).Further, Aerosols also affect the earth-atmospheric radiation budget directly in scattering and absorption of incoming solar radiation and indirectlyasclouds formation and precipitation (Ramachandran and Kedia, 2013). Thus, aerosols can influence the Indian monsoon (David et al., 2018). Earlier studies indicate that vehicular (Mahalakshmi et al., 2014, 2015) emissions, industrial, and thermal power plant emissions (Ramachandran et al., 2013) contribute significantly to atmospheric pollution, including gaseous pollutants. The ambient air quality is largely determined by the concentration of trace gases and particulate matter in the atmosphere (Nishanth et al., 2014). Increase in the concentration levels of trace gases and particulate matter has been a challenging environmental issue in urban and industrial areas.Numerous studies have been attempted across the globe to understand the air pollution concentrations during lockdown period and results indicate varied range of percentage reductions in pollutant concentrations. These studies are based on ground based measurements alone (Mohato and Ghosh, 2020;Mor et al., 2020) or satellite data alone (Biswal et al., 2020a; Xu et al., 2020) orwith a combinationof ground and satellite (Ratnam et al, 2020; Biswal et al., 2020b; Singh and Chauhan, 2020).Biswal et al. (2020a & 2020b) reported lockdown induced changes in tropospheric $NO_2$ variability over the urban and rural regions of Indian sub-continent with marked reduction of 30-50 % over



the urban and megacities. This change was mainly attributed to the reduced traffic emissions due to restriction on travel.In contrast to the above, increase inlevels of air pollutants during lockdown are also noticed at certain regions,which  are associated with natural emissions (dust storms, forest fires) and prevailing meteorological conditions. During India's phase-I of lockdown ($25^{th}$ March, 2020 to $7^{th}$ April, 2020), Ratnam et al. (2020) showed a decrease of $AOD_{550}$ over IGP region and a drastic increase over the central India, which were mainly

due to absence of anthropogenic activities and dominance of natural sources respectively.However the above said studieswere not performed detailed statistical analysis to indicate the observed changes are significantly lower than what could be expected due to inter-annual variability.

Objective of the present study is to understand the air quality quantitatively over the Indian region

under the control measures related to COVID-19 restrictions in the country. Thus, the present study examined the spatio-temporal variations of remotely sensed Tropospheric columnar $NO_2$ (TCN), Tropospheric columnar CO (TCC) and Aerosol Optical Depth ($AOD_{550}$)during LD and pre-LD and compared with preceding year (2019) and short-term mean (2014-2020). With these three air pollutants, we reported lockdown induced changes over the Indian region as a whole, hotspots (usual predominant sources) and urban regions along with

the statistical analysis(using de-trended data).However, no one attempted to study the TCC variability during lockdown over the Indian region using satellite measurements. To distinguish natural and anthropogenic emissions, we made an attempt to correlate the subsequent changes associated with meteorology, long range transport as well as forest fires.

## 2 Data

Satellite measured air pollutants data offer reliable, un-interrupted observations with high spatial and temporal coverage than ground-based measurements which are point observations.  Thus, the TCN observations from the Sentinel-5P/Tropospheric Monitoring Instrument (TROPOMI) and Aura/Ozone Monitoring Instrument (OMI), TCC data from high spatial resolution TROPOMI, $AOD_{550}$ data from Moderate Resolution Imaging Spectroradiometer (MODIS) Terra/Aqua sensors are used in the present study.  The brief details of these sensors

are given in Table 1. The TROPOMI was launched on $13^{th}$ October 2017 as the single payload on-board the Sentinel-5 Precursor (S5P) satellite of the European Space Agency (ESA). TROPOMI is a push-broom imaging spectrometer flying in sun-synchronous orbit at 824 km altitude and is designed to retrieve the concentration of several atmospheric constituents, which include TCN, TCC, $SO_2$ etc. It was developed jointly by ESA and Royal Netherlands Meteorological Institute (KNMI), which is the most advanced multispectral imaging spectrometer



(Martínez et al., 2020). OMI was successfully launched on National Aeronautics and Space Administration's (NASA's) Earth Observing System Aura Satellite which measures $NO_2$, $SO_2$ and aerosol characteristics. The MODIS sensor on-board NASA's the two Earth Observing System Terra and Aqua satellites are providing AOD retrievals.

| Parameter | Data source | Resolution | Website |
|---|---|---|---|
| TCN | Aura/OMI Sentinel-5P/TROPOMI | 0.25°×0.25° 3.5×7 $km^2$ (year, 2019) & 3.5×5.5 $km^2$ (year, 2020) | https://earthdata.nasa.gov/ |
| TCC | Sentinel-5P/TROPOMI | 7×7 $km^2$ (year, 2019) & 5.5×7$km^2$ (year, 2020) | https://earthdata.nasa.gov/ |
| AOD | MOD08_D3 from Terra & MYD08_D3 from Aqua | 1°×1° | https://ladsweb.modaps.eosdis.nasa.gov/ |
| Fire count | VIIRS | 375 m | https://firms.modaps.eosdis.nasa.gov/download/create.php |
| Winds and RH | ECMWF-ERA5 reanalysis | 0.25°×0.25° | https://cds.climate.copernicus.eu/cdsapp#!/dataset/reanalysis-era5-pressure-levels?tab=form |
| *Reference Period: Jan-July (2014 – 2020); Strict Lockdown Period: 25th March – 3rd May 2020* | | | |

**Table 1: Data resources**

Daily level 3 TCN data was obtained from Aura/OMI for computing short-term mean of TCN from 2015-2019. However, high spatial resolution TCN and TCC datafrom TROPOMI is used during LD periodof 2020 and equivalent period in 2019. The daily gridded global AOD product (Level 3) from the MODIS sensor on-board Terra (MOD08_D3_v6.1) Aerosol Optical Depth at 550 nm, Deep Blue algorithm, Land-only) and Aqua

(MYD08_D3_v6.1) Aerosol Optical Depth at 550 nm (Deep Blue algorithm, Land-only) satellites were used to investigate the aerosol loading over the Indian region for above said period. Detailed information about OMI sensor on-board Aura and MODIS sensor on-board Aqua/Terra as explained by Li et al. (2020). Over land, the previous studies reported that MODIS derived AOD uncertainty with respect to the Aerosol Network (AERONET) is ± 0.05 ± 0.20 × $AOD_{AERONET}$(Sayer et al., 2013;Levy et al. 2013). Details of MODIS AOD

retrieved algorithm for collection 6.1 and its validation can be found in Hsu et al., (2019) and Sayer et al.,



(2019) respectively. Inaddition to the above datasets, fire counts data from Visible Infrared Imaging Radiometer Suite (VIIRS)with confidence>80% were used. To understand the role of meteorology,winds from European Centre for Medium-Range Weather Forecasts (ECMWF) interim reanalysis which gives hourly data at different pressure levels (700 hPa and 850 hPa) was also used in the present study.Similarly, relative humidity from

ECMWF for the respective pressure levels is also used.

### 3. Methods

In the present study we attempted to assess the impact of lockdown on air quality over India by examining remotely sensed daily concentrations of TCN, TCC and $AOD_{550}$for the period of 01$^{st}$ January,2014 to October, 2020. Further, daily concentrations of the above said parameters were de-trended during the study

period to subside the inter-annual variability.Hence, phase wisechanges in TCN, TCC levels and $AOD_{550}$could be attributed to LD induced changes. Thus, the present study focused on the air pollution over the Indian region, its individual states, and state capitals during the strict lockdown period (phase-I and phase-II). Analysis of satellite-based observations of TCN from the years 2015 to 2020, TCC during 2019-2020 and $AOD_{550}$from 2014-2020 was carried out for lockdown period (phase-I and phase- II)as well as pre-lockdown period. Short-

term climatological means during pre-LD, phase-I, phase-II was computed for TCN from 2015 to 2020 and $AOD_{550}$from 2014 to 2020to assess the temporal changes of pollutants in the atmosphere. We have focused our analysis for the first two phases of lockdown in which the industrial and transport sectors were brought to a near standstill.

Figure 1 shows the data processing and execution strategy,which was followed in this study. The

detailed methodology used in this study is as follows. Python programming language is used to analyse TCN,TCC and $AOD_{550}$ variables during the study period as discussed in Figure 1. The parameters TCN, TCC and $AOD_{550}$ are extracted from the respective source files considering quality factors. Swath and mask are calculated for the region of Interest and the data is resampled to the required region of Interest using nearest neighbourhood algorithm.Further, time averaged maps of TCN, TCC and $AOD_{550}$for pre-lockdown, phase-I and

phase-II lockdown were generated for the years 2020 and 2019 along with difference maps.With respect to 2020, if the difference in pollutant concentrations (δx) is greater than zero indicating an increase effect and vice versa. Short-term climatological meansof TCN for the years 2015 - 2020and AOD for the years 2014-2020were computed. Thereafter, the regional increase/decrease in pollutant concentrations over the country and individual states were analysed.










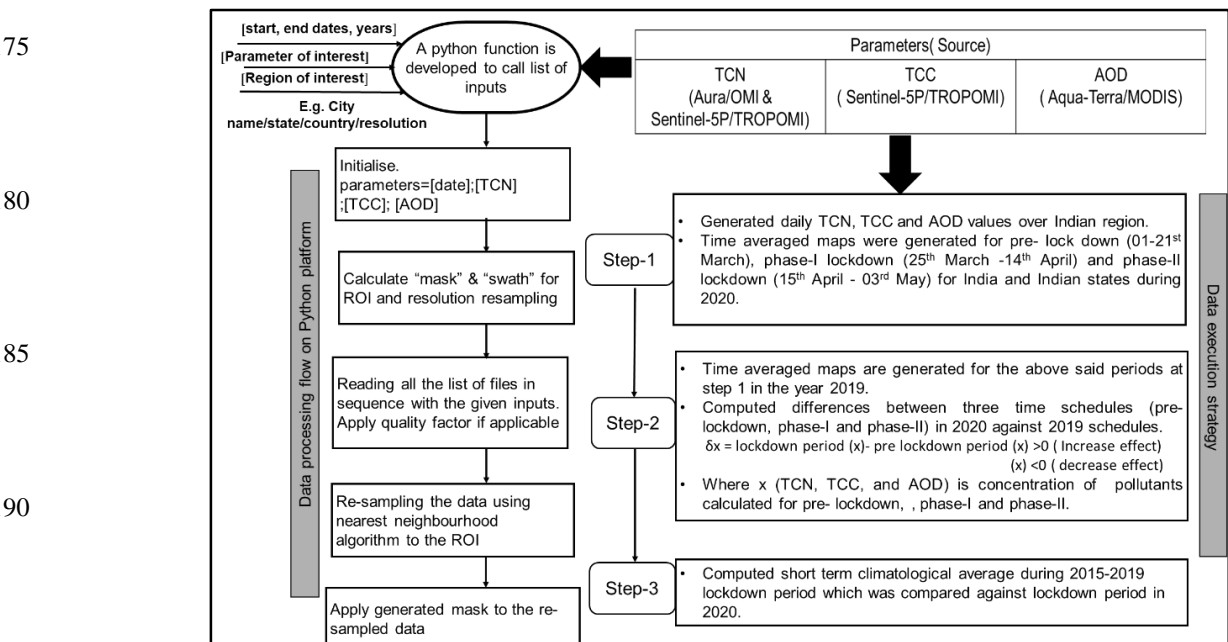

**Figure 1: Data processing steps and methodology**

### 3.1 Statistical metrics

Further, detailed statistical metrics for TCN, TCCand $AOD_{550}$ namely mean, standard deviation (SD), percentage of number of positive and negative pixels and student's paired t-test values were computed. To estimate metrics from the long-term data against lock period in 2020, here we utilized TCN data obtained from Aura/OMI and $AOD_{550}$ from MODIS. These metrics are calculated for every week starting from 1st January to 31st July as 31 weeks in total for the years 2015-2020 for TCN and 2014-2020 for AOD over the Indian region. Thus, the

following steps were implemented to quantify the metrics. These steps are writtenfor TCN as an example.

$$b_{2020-2019}(TCN) = TCN\ (2020) - TCN\ (2019) \qquad (1)$$

$$b_{2020-2015\ to\ 2019}(TCN) = TCN\ (2020) - TCN\ (2015-2019) \qquad (2)$$

$$1SD = \sqrt{\sum_{i=1}^{N} \frac{(b_i - \mu_b)^2}{N}} \qquad (3)$$



$$
\text{If } \left\{
\begin{array}{l}
b_i > 1SD, \text{ positive pixels } (P_p) \\
b_i < -1SD, \text{ negative pixels } (N_p) \\
-1SD \leq b_i \leq 1SD, \text{ neglect pixels}
\end{array}
\right. \tag{4}
$$

Where N is total number of qualified pixels over the Indian region and $\mu_b$ is mean bias. At each pixel, weekly bias (b) of TCN is estimated from the weekly mean TCN during 2020 lockdown period w.r.t 2019 and 2015-2019 period as shown in equations (1-2). For prominent change detection, additionally 1SD deviation filter was applied on the bias values of TCN and $AOD_{550}$. Therefore, if bias is greater than 1SD then the featuring pixels are classified as positive and if less than -1SD then they are considered as negative pixels. Pixels within ±1SD are omitted to avoid minimalistic feature changes and for better characterization. Subsequently, we computed the percentage of positive (Increased area) and negative pixels (decreased area) using following equations (5-6).

$$
\% \, P_p = \frac{\text{count}(P_p)}{N} \times 100 \tag{5}
$$

$$
\% \, N_p = \frac{\text{count}(N_p)}{N} \times 100 \tag{6}
$$

The same equations were applied on $AOD_{550}$ during 2014-2020 study periods. Further to understand LD induced changes in TCN and $AOD_{550}$ quantitatively daily mean values are de-trended using yearly data, which accounts for inter-annual variability in TCN and $AOD_{550}$ respectively. De-trended values of TCN and $AOD_{550}$ are generated by subtracting the linear regression estimated values from the daily means of TCN and $AOD_{550}$. To study the lockdown induced changes with significant levels, a paired t-test (Freedman et al., 2007) was implemented on the de-trended TCN and $AOD_{550}$ data during respective study period. The t-test follows a Student's t-distribution under the null hypothesis of $H_0$ with the means (μ) of two populations are equal ($\mu_1 = \mu_2$) with alternative hypothesis $H_a: \mu_1 \neq \mu_2$. To reject or accept the null hypothesis, p-value was used in this study. The hypothesis $H_0$ is rejected when a p-value is less than 0.05 and accepted if p-value is greater than 0.05 (5 % significance level).

## 4. Results and Discussion

Present study analysed the satellite based tropospheric columnar $NO_2$ (TCN), TCC and $AOD_{550}$ data to assess the lockdown induced changes over the Indian region.






### 4.1 Effect of Lockdown (LD) on TCN

The spatio-temporal variability in TCN concentrations during pre-lockdown and lockdown period (phase-I and phase-II) were analysed for the years 2019 and 2020 using high spatial resolution (Table 1) Sentinel-5P/TROPOMI. Temporally averaged concentrations of TCNduringpre-LD (01st March to 21st March 2020),

phase-I and phase-II of lockdown, and the corresponding period of the previous year (2019) are shown in Figures 2a-c, along with the differences in concentration levels between different periods.During pre-LD time of years 2019 and 2020 as shown in Figure 2a, extent of TCN hotspot regions (majorly Eastern and National Capital region) remain same however, a mild reduction of TCN noticed during pre-LD of 2020 compared to 2019. This could be due to inter-annual variability in TCN levels and also the absence of source emissions due

to lockdown imposed by neighbouring countries via long-range transport (For example: lockdown imposed in China from 23rd Jan, 2020, Italy from 21st Feb 2020 and Malaysia from 18th March 2020).

The mean TCN over the entire country during phase-I of 2020 and 2019 are $1.53 \times 10^{15}$ and $1.86 \times 10^{15}$ molecules cm$^{-2}$ respectively. A reduction of about 22 % TCN levels are observed in 2020 compared to 2019 during phase-I, which accounts both inter-annual variability and lockdown-imposed changes. Further to

understand the lockdown induced changesin TCN levels, study was focused on TCN hotspots, which includes power plants and metropolitan cities with industrial and transport activities. To contain the spread of COVID-19, about 95 % of the anthropogenic activities were halted (Ratnam et al., 2020) during phase-I LD. However, during phase-Iafew anthropogenic emissions are present underessential services (pharma industries, power plants, medical services, vehicles that were carrying daily commodities) and indoor emissions. As a result, the

TCN levels and its area of extent as shown in Figure 2b during phase-I of LD in 2020 over the hotspot regions (Eastern region of India) decreased by 22 % when compared to equivalent period of 2019. The eastern region of India has a significant number of major power plants and refineries with associated industries. During phase-I LD in 2020, a reduction of TCN levels is observed over this region, which is due to shutdown of industries and urban activities such as transport and small-scale industries. However, country's high TCN levels are noticed in

the eastern region with less spread indicating the active role of power plant industries located in this region. Simultaneously, the National capital region (NCR) shows marked reduction by about ~70 % during phase-I LD. The major source of emissions in the NCR are dominated by heavy traffic, densely located industries, and industries of steel, cement and sugar (Ghude et al., 2008). As mentioned above, India's strict lockdown permitted only essential services therefore remaining all activities were halted during phase-I period. Thus,





resulted in low levels of TCN over the hotspot regions except in eastern region due to continuous operation of
power plants (Biswal et al., 2020b).

To sustain the Indian economy, activities namely agricultural practices and associated activities (crop
residue burning) are given permission during phase-II along with phase-I restrictions. Figure 2c shows TCN
levels and their difference map during phase-II LD.With respect to the same period of phase-II in 2019, the

TCN levels over the country decreased by 13 % with mean TCN of $1.55\times10^{15}$ molecules cm$^{-2}$and $1.75\times10^{15}$
molecules cm$^{-2}$ in 2020 and 2019. Thus, continued decrease of TCN levels are recorded over the hotspot
regions. However, an increase of TCN also observed over the neighbouring regions of eastern region clearly
indicating the dispersion of TCN. In contrast to earlier, an increase in TCN levels over north-east region could
be due to seasonal biomass burning in this region.Thus, the mean TCN levels over the entire country is

$1.54\times10^{15}$ molecules cm$^{-2}$during total LD (phase-I and phase-II together) with a reduction of 18 % compared to
respective period in 2019 as well as with respect to 5 years mean TCN .

Overall, southern part of India reported less TCN values as compared to eastern and NCR regions
(hotspot regions) during pre-LD, phase-I and Phase-II. The hotspots over southern part of India are not as dense
as in the eastern and northern regionscould be one of the reasons for its lower values. Theother reason could be

dispersion of TCN values as result ofland and sea breeze effect since it is surrounded by Bay of Bengal and
Arabian Seaover this region (Ramachandran et al., 2013). Thus, reduced TCN levels over southern part of India
irrespective of LD were observed due to above facts. Weekly variations of TCN were also shown in Figure 2d to
assess the extent of source emission during the lockdown period.Therefore, present study depicted the possible
driving factors of TCN values during pre-LD, phase-I and phase-II using high resolution spatial data from

satellite.





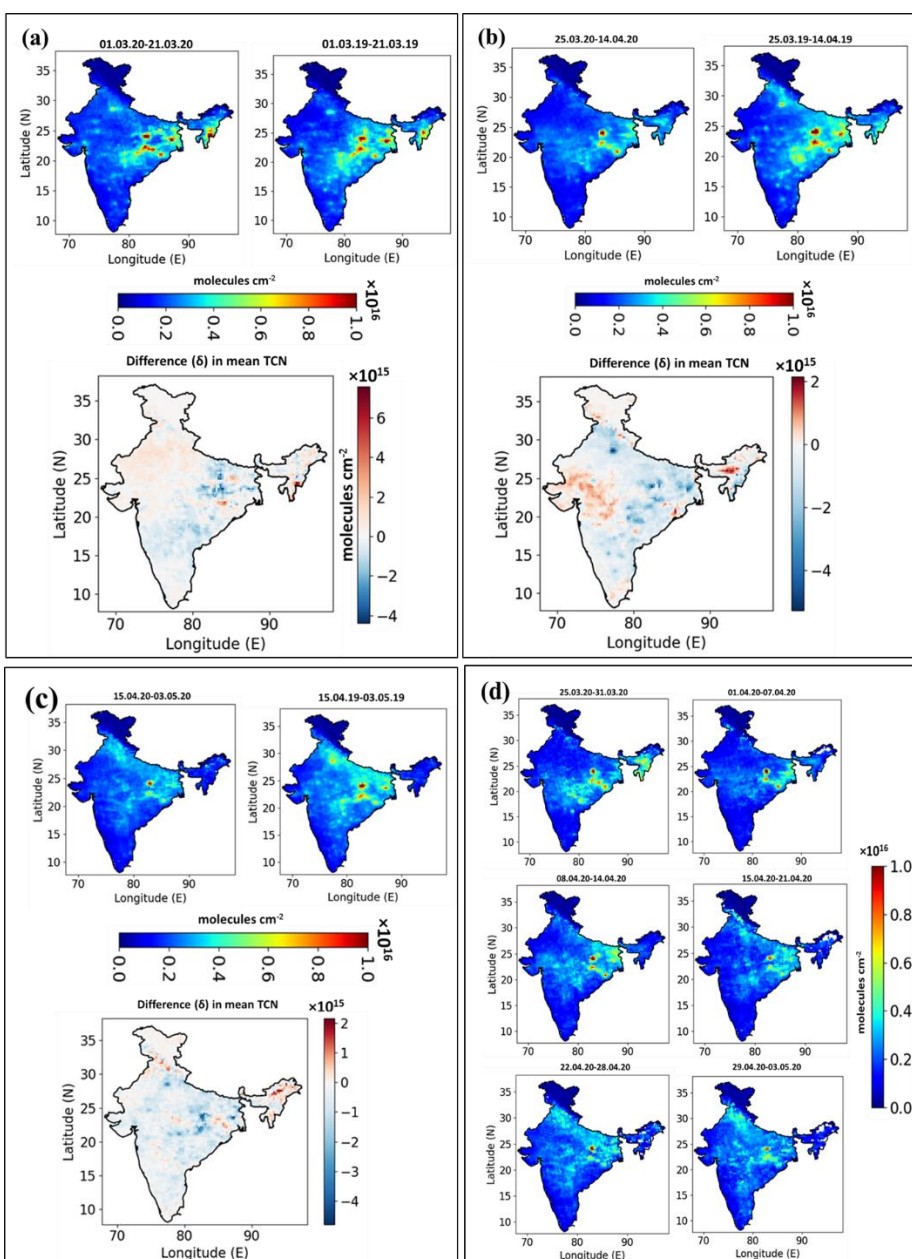

**Figure 2: Time-averaged TROPOMI TCN concentration and their difference maps between 2020 and 2019 during a)pre-LD b) phase-I lockdown c) phase-II lockdown, d) weekly mean TCN variation starting from 25th March to 03rd May 2020.**





### 4.1.1 Short-term climatological variations in TCN due to lockdown

Time series analysis of TCN was carried out during 2015-2020 for the months of January to July over the entire Indian region covering cold and hotspot regions as shown in Figures3a-d. A smoothing function with span of 7 days was used for better visualization of patterns/trends in TCN levels as shown in Figures3a-b, with red (green) bars indicating increase (decrease) TCN levels in 2020 relative to 2019 and mean of 2015-2019. The 7-day moving average show significant decreaseof TCN concentration with 99.99% (p-value $<<$ 0.05) confidence

interval during total lockdown period.However, it is also noticed that a decrease of TCN during prior and post lockdown periods, which is further tested statistically and found insignificant change with a p-values of 0.08 and 0.24 respectively. Further statistical significance of TCN variability across hotspot, cold spot regions and also over the major cities where TCN dropped (↓) drastically (except NE) along their percent drop during total LD when compared to 5 years mean TCN levers were summarized in Table 2. It clearly shows the TCN levels over

the IGP (22% ↓), eastern region (29% ↓), and major cities (New Delhi 54% ↓) declinedsignificantly compared to preceding 5 years mean TCN levels during the total LD period. Change in TCN during the study period is also associated with the inter-annual and seasonal variability besides its dominant anthropogenic sources. Figure 3c shows annual means of TCN in pre-LD, phase-I and phase-II LD during 2015-2020 period.It clearly depicts inter-annual variability in TCN between the years at each time scale along with the lockdown-imposed changes.

Between the time scales during the study period, a clear seasonality in TCN levels is also observed in Figure 3c.

       The horizontal bar plots in Figure 3d showsthe rate of change (RoC) in TCN levels in 2020 against the mean during 2015-2019 indicating the impact of lockdown on TCN concentrations over Indian region.The RoC is extremely important in weather and climatological studies because it allows understanding and predicting the

trends/patterns in climatic parameters. RoC is used to describe the percentage change in a parameter over a defined time period and it represents the rate of acceleration of the parameter. To compute the RoC in this study, we have used de-trended TCN daily values, which accounts inter-annual variability in TCN.Thus, the present RoC depicts the TCN variability due to the lockdown-imposed changes alone. There is a clearly observable lowering in TCN levels relative to the short-term climatological mean by 12% for the pre-lockdown period as

Lal et al. (2020) also reported similar results. During period from January-April, 2020, authors reported a substantial reduction in the level of TCN, TCC, and $AOD_{550}$ across the globe during COVID-19 pandemic as each country (at different spatial scale) imposed lockdown at different time scales. This could be the reason for reduction in concentrations of TCN during pre-LD period in 2020 as compared with 2019 as well as mean





picture of 2015-2019. Furthermore, a noteworthy reduction of TCN concentration by 18% and 15% is observed

for the phase-I and phase-II lockdowns respectively over the Indian region.

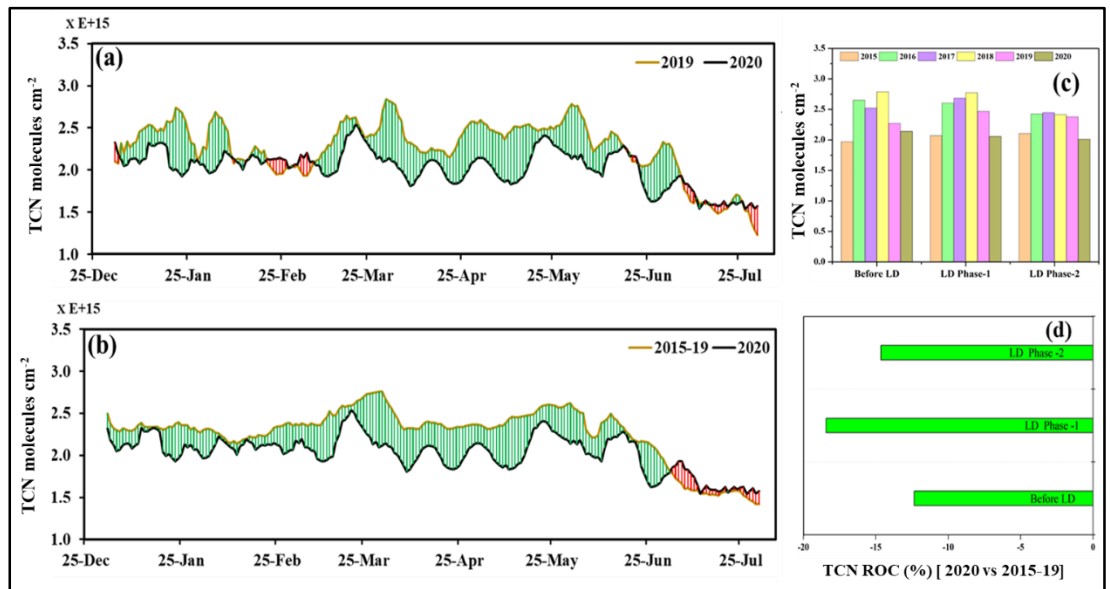

**Figure 3: Moving average time series analysis during January-July of TCN during a) 2019 vs. 2020; b) short-term climatological mean of TCN (2015-2019) vs. 2020  c) Annual variations of TCN (2015-2020) during period of before lockdown and different phases of lockdown  and d) Rate of change of TCN during 2020 w.r.t. 2015-2019 period.**

| Region/City | *Student's Paired t-test p-value (RoC in percent during Total LD) | | |
|---|---|---|---|
| | Pre-LD | During total LD | Post LD |
| IGP | 0.03 | << 0.05 (22 % ↓) | 0.31 |
| East | 0.62 | << 0.05 (29 % ↓) | 0.11 |
| NE | 0.66 | 0.19 (3 % ↑) | 0.55 |
| New Delhi | 0.57 | 0.0002 (54% ↓) | 0.05 |
| Bangalore | 0.58 | $2.62E^{-5}$(43%↓) | 0.17 |
| Chennai | 0.37 | 0.012 (41%↓) | <<0.05 |
| Mumbai | 0.95 | 0.011 (35%↓) | 0.17 |
| Hyderabad | 0.49 | 0.0003 (30%↓) | 0.007 |
| *p-value < 0.05 is significant and vice-versa; (↓, ↑) indicates (decrease, increase) | | | |

**Table 2: Student's paired t-test for TCN during the lockdown period against 5-year mean (2015-2019)**



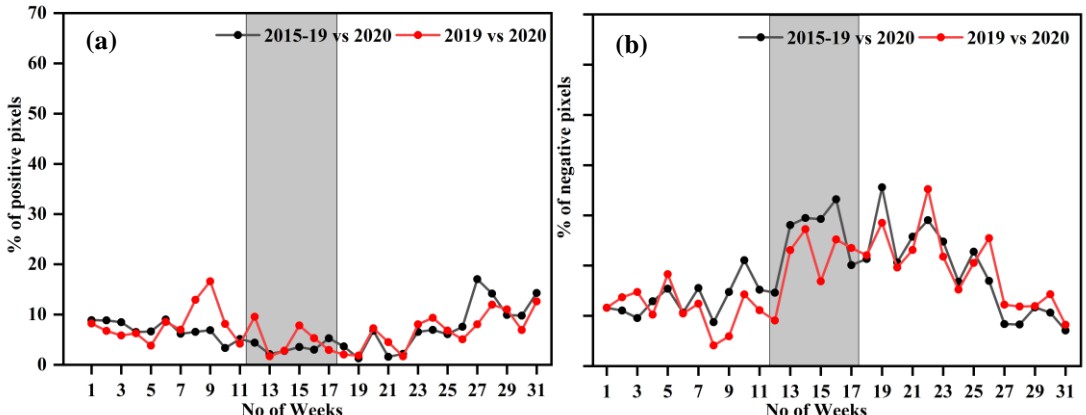

**Figure4: Aura/OMI measured TCNa) percentage of positive pixels b) percentage of negative pixels during**
360     **the period 2020 vs (2015-19) and 2020 vs 2019.**

Figures4a-b show statistically computedTCN metrics for number of positive pixels and number ofnegative
pixels in percentage during January to July period as number of weeks which starts from 01st January. Weekly
TCN means for the years 2015-2019, 2019 and 2020 were used to calculate positive and negative pixel count
365     based on methodology stated in section 3.1. Thus, red color line in the Figures 4a-b represent the number of
positive/negative pixels for the years 2019vs.2020 and black color line represents the same for the years 2015-
2019 vs 2020. The study showed a greaternumber of negative pixels (decreased area) during lockdown weeks
and vice-versafor positive pixels which clearly depicts the extent of area affected due to LD and subsequent
changes on air pollutants over Indian region.


**4.2 Effect of LD on TCC**

 The mean TCC levels over Indian region during the pre-lockdown and LD periods were studied to assess the
COVID-19 lockdown induced changes in TCC. Figures 5a-c show mean concentrations of TCC over the Indian
region during pre-LD, phase-I and phase-II LD periods using TROPOMI data. During the pre-lockdown period
(01-21st March 2020), TCC levels were higher (mean = $2.39 \times 10^{18}$ molecules $cm^{-2}$) as compared to 2019 by
~4.8%, which indicates increasing effect of anthropogenic activitiesand inter-annual variability. As shown in





Figure 5b difference map, the TCC levels increased in north-eastern (NE) region followedby part of central India (CI) and south of north-west (S-NW) of India as compared to 2019 of phase-I LD period. An observed

increase over these regions was evaluated statistically and found insignificant (p >0.05).An increase of TCC in NE region of India is mainly attributed to the active fire counts (Figure 5d) during the phase-I of LD as shown in Figure 5b. During phase-I, other regions of India namely the Indo Gangetic plain (IGP), north and south regions show decreased TCC levels compared to same period of 2019. The decreased TCC levels in these regionsduring phase-I are attributed to the shutdown of industries (cement, sugar and steel etc.), absence of transportation and

restriction on crop residue burning. However, household emissions due to residential cooking are still present during lockdown which is major contribution to CO from rural areas and some parts of urban region (slums).In India, 72 % of the populations live in rural and urban slums and most of them are continued to usehousehold biofuel for cookingunder lockdown (Verma et al., 2018; Beig et al., 2020).

        However, the mean TCC levels as shown in Figure 5c are higher during the phase-II of lockdown. Over

the entire country, the mean TCC value during phase-II is 2.38 $\times 10^{18}$ molecules cm$^{-2}$ in comparison to 2019 mean value of 2.32$\times 10^{18}$ molecules cm$^{-2}$. In phase-II of LD, the TCC levels are decreased in NE region, which is strongly attributed to the reduced active fire activity in this region as shown in Figure 5c. Except in NE region, consistent increase of TCC levels is observed during phase-II. Since agriculture farming industry is exempted in the phase-II LD and observed active fire counts in the central India, thus observed enhancement in

the TCC levels during phase-II. An increase or decrease of TCC levels in the atmosphere is mainly dominated significantly by anthropogenicactivitiescompared to natural emissions (Kanchana et al., 2020) as discussed earlier. However, comprehensive reasons for the increase of TCC levels in phase-II are not investigated in this study.











**Figure 5:**Sentinel-5P/TROPOMI derivedTime averaged TCC concentration and their difference maps in 2020 and 2019 a) pre-LD b) phase-I lockdown c) phase-II lockdown, d)Fire counts from VIIRS for phase-I, phase-II during 2020.





### 4.3 Effect of LD on AOD$_{550}$

We have used Terra-Aqua/MODIS derived AOD$_{550}$during 2014-2020 for the months of January to July to understand the lockdown-imposed changes. Terra/MODIS AOD$_{550}$ represents the footprint for 10:30 and Aqua/MODIS AOD$_{550}$for the 13:30 local time. As we observed similar spatial variation of AOD$_{550}$ from both Terra-Aqua/MODIS, only Aqua/MODIS derived AOD$_{550}$ is shown here (Figure 6). AOD$_{550}$ levels over the Indian region for 2019, 2020 and the difference in AOD$_{550}$ for both years for pre-lockdown period is depicted in Figure 6a. During this period, the AOD$_{550}$ levels for 2020 over the IGP region (~21% of the Indian Territory landmass) is more compared to rest of the regions ofIndia which is expected throughout the year. This is mainly because of its orographic effect and densely populated (accommodating ~40% of the Indian population). The main anthropogenic sources over IGP region arecoal-based power plants and industries, crop residue and forest fires and household cooking which contribute to high AOD in this region.Thus, the IGP is known as first hotspot for anthropogenic aerosol emission in South Asia.During phase-I of LD as shown in Figure 6b, aerosol loading over the IGP region attained its baseline concentration (~45% dropw.r.t.  2019 of the same period) due to the strict implementation of LD. This region is densely populated and congested industrial activities, which were shut down during this period resulted a nearly AOD free atmosphere. This indicates absence of anthropogenic activities due to mobility restrictions. Further, prevailing meteorology over IGP (high wind speed and low relative humidity at 850 hPa and 700 hPa)is also favoured for decrease in AOD$_{550}$ during phase-I LD.

Despite the strict LD in the country, unexpected increase in AOD$_{550}$ is observed by ~28 % compared to preceding year of the same period over Central India (CI) which ispredominantly dominated by dust storms (Ratnam et al., 2020) through long range transport and prevailing meteorology (Pandey et al., 2020). Thus, to understand the prevailing meteorology over CI,phase wise relativehumidity and wind speed at pressure levels 850 hPa and 700 hPa respectively were analysed as shown in Figures 7a-b. During phase-I and phase-II, majority of the winds over CI dominated by westerly (calm winds) with high relative humidity. Under this prevailing meteorology, calm winds contribute to slow dispersion and high RH modulates the aerosol chemistry and hygroscopic growth mechanism (Pandey et al., 2020). As a result, the increase of AOD$_{550}$over CI is observed.Further, high AOD$_{550}$ over NE regions also observed because ofhigh active forest fire counts(Figure 5d)compared to 2019 LD period.Figure 6c shows AOD$_{550}$ during phase-II of India's LD in 2020 against AOD$_{550}$ in 2019 of same period.During this phase, an increase inAOD$_{550}$ (~ 3%)over IGP was observed. Over CI, a reduction of AOD$_{550}$(~ 18%) was observed compared to phase-I of LD andnot much change (~1%)when compared to respective period in 2019which clearly depicts reversal of meteorology in phase-II with respect to





470    phase-I. Causative factors for this decrease over CI w.r.t phase-I are due to low RH and high wind speed at 700

hPa and 850 hPa over this region.

In a Nutshell, this study demonstrates the lockdown induced Terra/MODIS $AOD_{550}$ changes over the

IGP and CI during total LD period shows a significant change with p-value of 0.01 (99 % confidence

interval)with a decrease of  20 % over IGP and 0.03 (97 % confidence interval)with an increase of 25 % over CI

475    when compared to equivalent period in 2019.

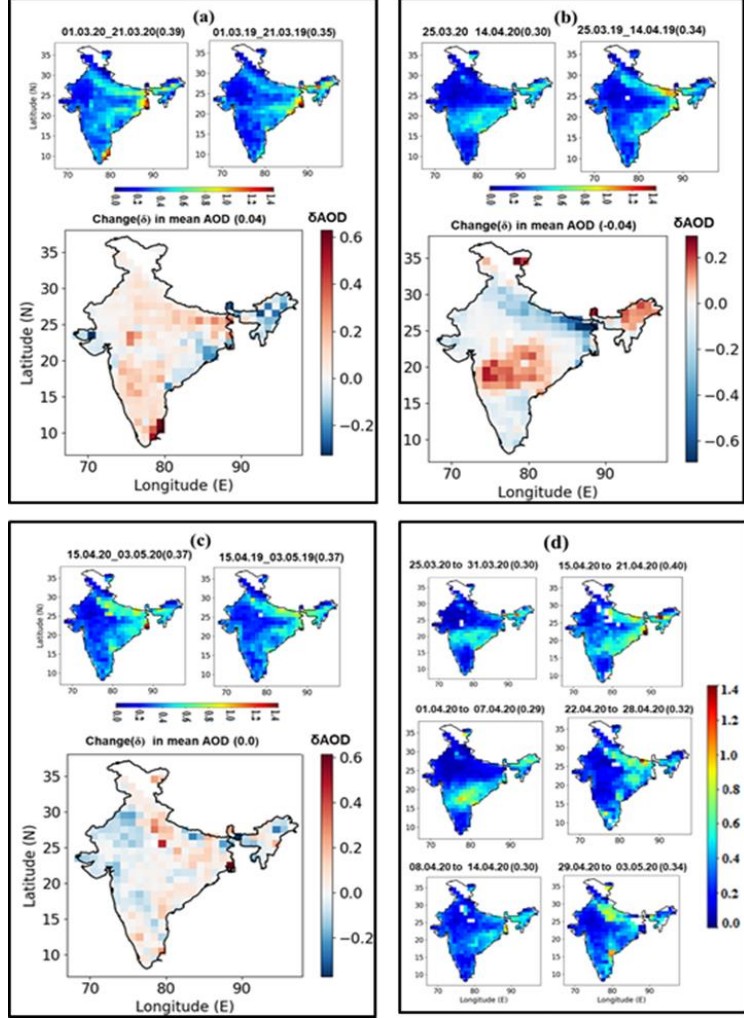

**Figure 6:Aqua /MODIS derived Time averaged $AOD_{550}$ and their difference maps in 2020 and 2019 a)**

**pre-LD b) phase-I lockdown c) phase-II lockdown d) weekly variation in total lock period during 2020.**

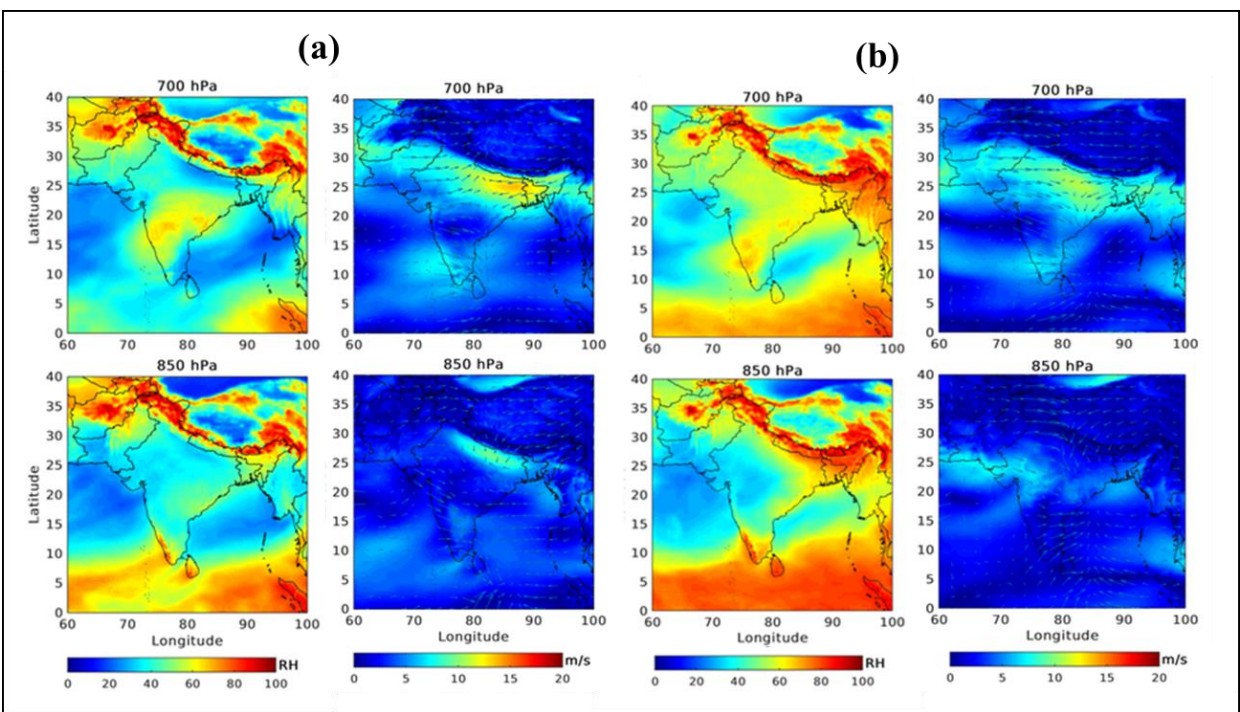

**Figure 7: Mean relative humidity (%) and mean winds (ms⁻¹) observed at 700 hPa and 850 hPa (a) phase-I of lockdown and (b) phase-II of lockdown**


### 4.3.1 Short-term climatological variation of AOD$_{550}$ due to lockdown

Aerosol optical depth is one of the important short-term climatic forcing agents along with long lived greenhouse gases namely carbon dioxide ($CO_2$), methane ($CH_4$), water vapor ($H_2O$) and nitrous oxide ($N_2O$). A 7-day smoothing average filter was applied on AOD$_{550}$ time series data as discussed in section 4.1.1. Figures 8a-

d show 7-day moving average time series analysis of AOD$_{550}$ levels for MODIS Terra and Aqua from January to July over the Indian region for 2014-2019 mean values, 2019 and 2020. AOD$_{550}$ measured by Terra/Aqua-MODIS (Figures 8b & 8d) show a significant change in aerosol loading over the country during the lockdown period in 2020 compared to mean AOD$_{550}$of 2014-2019. Statistical analysis of Student's paired t-test shows a strong significant change in AOD$_{550}$ with a p-value of $<<0.05$ for Terra/Aqua-MODIS during the total LD

against 6-year mean (2014-2019). Interestingly the present analysis show a very much significant change of AOD$_{550}$ during post LD compared to LD with p-value $<<<0.05$ (order of an Integer$\times10^{-12}$), which attributes to





continued effect of lockdownas phase-III and IV  and scavenging effect during monsoon season. Due to increase of precipitation in the active summer monsoon (June-July) season, lowering of aerosols is expected (Boucher et al., 2013). Thus, the continued lockdown and active monsoon improved the air quality beyond strict lockdown period as shown in Figures8a-d.


The annual mean $AOD_{550}$over the Indian region in each phase is shown as vertical bars in Figures8e-f, indicating the inter-annual variability of $AOD_{550}$ across the phases and seasonal modulation between the phases during 2014-2020. Despite inter-annual and seasonal variability of $AOD_{550}$, the strict lockdown in 2020 showsa decrease in phase-I and phase-II compared to pre-LD, which could be associated with the reduced anthropogenic sources besides prevailing meteorology as discussed at section 4.3.The RoC in $AOD_{550}$ was computed (Figures 8e-f) to understand the effect of short-term climatological mean $AOD_{550}$ over lockdown period in 2020. A positive RoC of +8.8 % (+14%) was observed during pre-LD as measured by Terra/MODIS (Aqua/MODIS) against 6-year mean $AOD_{550}$. This increase is tested statistically and found insignificant with p-values of 0.11 and 0.37 for Terra/MODIS and Aqua/MODIS respectively. During phase-I (phase-II) Terra /MODIS showed statistically significant negative RoC with -24 % (- 9%) and Aqua/MODIS showed -22 % (-7%) against 6-year mean $AOD_{550}$as most of the sectors were turned off except household emissions and essential services.Therefore, this study reports, India's strict lockdown improved the aerosol air quality over the country with markedly changes over the IGP and CI respectively.








**Figure 8: a) Moving average time series analysis of $AOD_{550}$ measured by Terra/MODIS during 2019 and 2020; b) Terra/MODIS short-term climatological mean of $AOD_{550}$ (2014-2019) vs. 2020; c) time series $AOD_{550}$ measured by Aqua/MODIS during 2019 and 2020; d) Aqua/MODIS short-term climatological mean of $AOD_{550}$ (2014-2019) vs. 2020; e) Variations of Terra/MODIS measured $AOD_{550}$ before Lockdown and different phases of Lockdown and respective RoC; f) Variations of Aqua/MODIS measured $AOD_{550}$ before Lockdown and different phases of Lockdown and respective RoC.**






Figures 9a-d show number of positive and negative AOD$_{550}$pixels in percentage at weekly interval computed from the respective biases during the study periodover the Indian region. Figures9a-b show the percentage of positive and negative pixels of AOD$_{550}$ measured by the Terra/MODIS. During the lockdown weeks (shaded in grey color) in 2020, the number of positive pixels were less w.r.t 2019 and short-term climatological mean of

AOD. Figure 9b shows more percentage of negative pixel during the same study period indicating the larger area of extent with lower AOD$_{550}$ due to strict lockdown in India. This change is even high w.r.t short-term climatological mean of AOD$_{550}$. The Aqua/MODIS derived AOD$_{550}$ also shows similar variability and is as shown in Figures 9c-d.




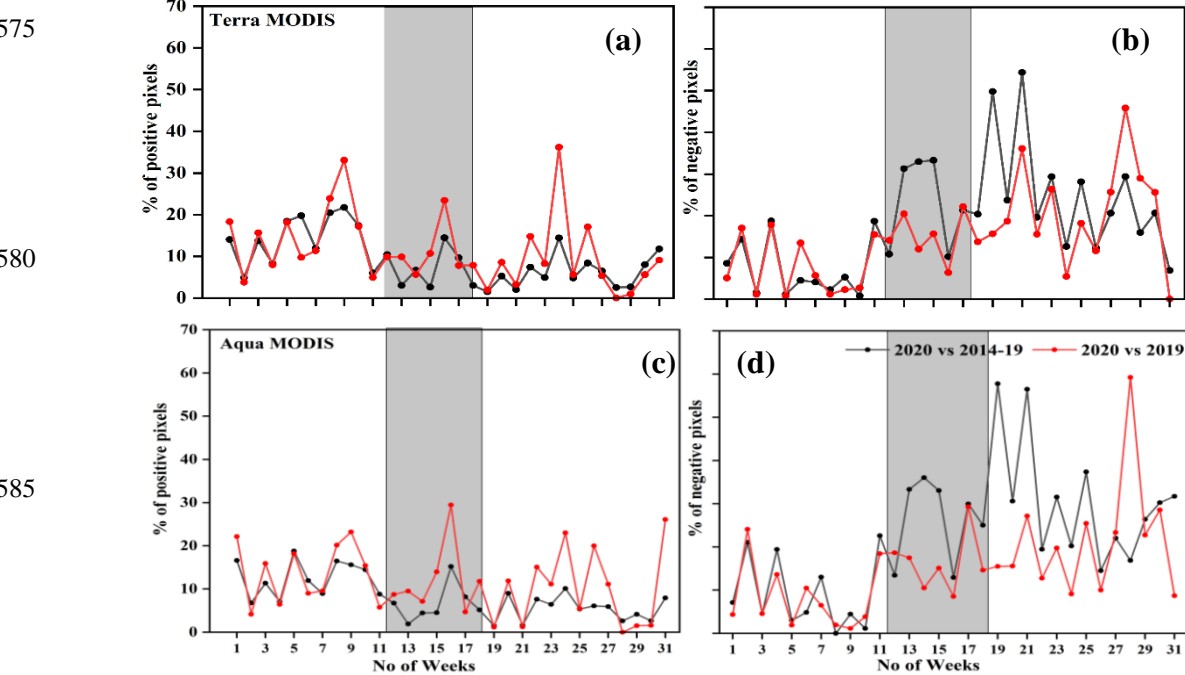


**Figure 9:Terra/MODIS(a) percentageof number of positive pixels b) percentage of number of negative pixels during the period 2020 vs (2014-19) and 2020 vs 2019.Aqua/MODIS (c) percentage of positive pixels d) percentage of negative pixels during the period 2020 vs (2014-19) and 2020 vs 2019.**






**4.4 State wise Rate of Change (RoC) of TCN w.r.t. 2015-2019 and AOD w.r.t. 2014-2019**

Figures 10a-c show state-wise RoCcomputed for pre-LD, phase-I, phase II and total LD phases in 2020 with
respect to 5 years mean (2015-2019) for TCN and with respect to 6 years mean (2014-2019) for Terra/AOD$_{550}$
and Aqua/ AOD$_{550}$. Positive percentage (dark blue above zero) of RoCindicates increase of pollutants for the
respective phases shown in the Figure 10 when compared to same phase period of means of TCN for 2015-2019
and means of AOD$_{550}$ for 2014-2019. Negative percentage (light blue below zero) of RoC indicates decrease of
TCN and AOD$_{550}$ w.r.t 2015-2019 and 2014-2019 values respectively for the phases shown in the same Figure
10. Results clearly depict the change of pollutants over each state during the lockdown period compared to
respective period in 5 years mean for TCN and 6 years mean for both AOD$_{550}$.

During the total lockdown period in 2020 w.r.t the mean of TCN (Top right in Figure 10a) during the
same period for 2015-2019, the TCN values are clearly dropped in the hotspot zones namely eastern states
(Odisha, Chhatisgarh and Jharkhand) and NCR regions (New Delhi, Ghaziabad, Faridabad, Gurugram and
Noida).Thus, the drop in TCN values over these regions are evaluated statistically and found significant change.
In similar manner, the AOD$_{550}$ measured by the Terra-Aqua/MODIS also shows strong reduction over IGP
region during the total LD (top right Figures 10b-c). However, unexpected increasing effect is noticed in the CI
states with respect to 6 years mean of respect AOD$_{550}$ during phase-I. Similar results are also observed when
compared to preceding (2019) year mean AOD$_{550}$ which is discussed earlier section in detail manner. Further, it
is observed that, the negative RoC of AOD$_{550}$ over IGP region during phase-I is more prominent compared to
phase-II RoC.It is also noticed further that the RoC of AOD$_{550}$ computed from the Terra-Aqua/MODIS showing
similar trends during the total lockdown period with small difference in the amplitudes. This difference of
amplitude between these two sensors could be due to difference in overpass time, which changes atmospheric
dynamics such planetary boundary layer height, solar zenith angle and prevailing meteorology. An average of
Terra/Aqua MODIS derived RoC of AOD$_{550}$ show strong reduction in the western part of India mainly
Rajasthan (-36 %) and Gujarat (-31 %) respectively during the total LD period(Ranjan et al., 2020). Therefore,
in a nutshell an analysis of RoC depicts regional variability of air pollutants during the total LD period in 2020
w.r.t to short-term (5-6 years) mean.


**Figure 10: State wise RoC computed for pre-LD, phase-I, phase-II and total LD a) TCN b) Terra/MODIS derived AOD$_{550}$ and c) Aqua/MODIS derived AOD$_{550}$**





## 5 Conclusions

The present study was carried out an analysis on air pollution in connection with the world's largest lockdown imposed by Government of India to contain the spread of COVID-19. The lockdown was extended as 4 lockdowns with strict lockdown from the phase-I to several relaxations in the phase-IV. However, the lockdown was near total only in phase-I and II, with the total shutdown of industrial and transport sectors. Thus, we have only considered first two phases in the present studyas total lockdown. We used satellite-based observations of

tropospheric TCN, TCC and $AOD_{550}$ pollutant concentrations analysed during the period of lockdown and prior to LD against the same period of the preceding year (2019) and also against the short-term mean (2014-2019) for about 6 years.

Following are the major findings from the present study

- Due to India's strict LD, the TCN levels are dropped significantly to 18 % across the country compared

to preceding year with a p-value of 0.0007(confidence interval of 99.93 %).

- Further, analysis is emphasised over the TCN hotspot regions of the Indian sub-continent and observed reduction of (29%) TCN during the total LD period with higher confidence interval.

- The TCN levels with respect to short-term climatological mean are markedly dropped over the urban locations namely New Delhi (-54%), Bangalore (-43 %), Chennai (-41 %), Mumbai (-35 %) and

Hyderabad (-30 %) respectively with high confidence interval about 99.90 %.

- However, during the total LD, an unexpected increase of TCN levels are recorded over NE region, which is directly correlated with the seasonal biomass burning in this region. This increase is also evaluated statistically against 5-year mean TCN and found insignificant with p-value of 0.19.

- The TCC levels are decreased during the phase-I over IGP, north and south regions which could be due

to the absence of transportation and shutdown of industries. Although, variability in the TCC levels werenoticed during the total LD period it was tested statistically and found insignificant.Observed high tropospheric CO levels in the NE region during phase-I LD period, which is mainly attributed to the active fire counts in this region. Also observed low TCC levels in the NE region during phase-II due to the diminished effect of fire counts.

- Since IGP region is densely populated and clustered industries, which were shut down during phase-I of India's LD, the $AOD_{550}$ levels are attained to near baseline in this region (AOD mean value=0.2). This drastic decrease of $AOD_{550}$ in the IGP region statistically evaluated and found very significant with a p-value of 0.008 with preceding year (45% decrease) and 50 % reduction against 6-year mean with a p-value $\ll 0.05$.





- Despite the country's LD, the $AOD_{550}$ levels are high over the CI, which were predominantly dominated by the transportation of dust storms and prevailing meteorology. Also observed high $AOD_{550}$ over NE and is associated with active fire counts. However, this increase is significant in the CI with a p-value 0.03 and insignificant in the NE region with a p-value of 0.33, which indicates insignificant change due to LD.

- The LD induced changes in $AOD_{550}$ measured by the Terra-Aqua/MODIS show a significant change over the Indian region with very high confidence against 6-year short-term climatological mean. This variability helps to improve the regional air quality.

- Further, an analysis of RoCwas carried out to depict the regional variability of air pollutants during the total LD period in 2020 w.r.t to short-term climatological mean.

Therefore, this study successfully demonstrates the satellite based TCN, TCC and $AOD_{550}$ changes due to the India's lockdown during 2020 and compared against preceding year (2019) and also against the short-term mean picture of 2014-2019.

**6 Code/Data Availability**

The satellite and reanalysis data used in the present study are freely available and can be downloaded as summarized in Table 1 with user's credentials.

**7 Author Contribution**

Conceptualization and Formal analysis are done by MP,AM, SH, DVM,VKS; Writing – original draft, MP, DVM, ALK; Writing – review & editing DVM, ALK, JS, SSR, MVR and UV

**8 Conflicts of Interest**

The authors declare no conflict of interest.



## Acknowledgement

Authors sincerely thank Dr. Raj Kumar, Director NRSC for his support and encouragement for carrying out this study. We greatly acknowledge Earth data web portal for providing the free access to the Aura/OMI, Sentinel-5P/TROPOMI satellites data. We also greatly acknowledge LAADS (Level-1 and Atmosphere Archive and Distribution system) operated by National Aeronautics and Space Administration (NASA) for providing Aqua-Terra/MODIS satellite data. Authors would also like to thank Land, Atmosphere Near real-time capability for Earth Observation system (LANCE)/ Fire Information for Resource Management System (FIRMS) operated by the NASA for providing the fire data. Authors further thank European Centre for Medium Range weather Forecasts (ECMWF) for providing the Relative humidity and wind data. We thank Dr. P. Raja, Principle Scientist, Indian Institute of Soil and Water Conservation-Indian Council of Agriculture Research, Ooty, India for reviewing the manuscript.

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
