# Peer review of "Measurement report: An assessment of the impact of a nationwide lockdown on air pollution – a remote sensing perspective over India"

_Atmospheric Chemistry and Physics, 2020_

## Author Response (AR1)

Here authors use tropospheric $NO_2$, CO and $AOD_{550}$ nm observations from OMI, TROPOMI and MODIS satellite instruments to quantify changes in atmospheric composition during different phases on lockdown in India. Authors show that composition changes are non-uniform over India and some region even indicate increase in tropospheric $NO_2$ column or $AOD_{550}$nm, that could be attributed in the changes in fire activity or changes in meteorological fields. However, there are still significant shortcomings in the manuscript. I would recommend the manuscript for publication, if authors can address some of the concerns mentioned below.

Reply: Thanks for the comments and suggestions, which certainly helped us to improve the manuscript further with the following comments. As advised, we attempted to incorporate the following line by line suggestions.

Major Comments:

Authors did not use simple spell check available in Microsoft word. It is not easy to –review a manuscript with so many spelling mistakes and/or forgotten spaces.

Reply: Thank you for the suggestion. The revised manuscript is carefully checked and the spelling mistakes were corrected accordingly.

Data processing section is very vague. I could not understand which data version is used, how level 2 data was converted in level 3, or which data quality flags are used. It would be good idea to write in detail (or provide Python code in the Appendix). I will also strongly recommend authors to upload processed data file on some online repository such as https://zenodo.org, so that interested readers can compare it against their processing method.

Reply: Thank you for the suggestion. In the present study regarding data version, we didn't convert any level 2 data to level 3. Daily level 3 tropospheric columnar $NO_2$ (TCN) data was obtained from Aura/Ozone Monitoring Instrument for computing short-term climatological mean (2015-2019). Level 2 tropospheric $NO_2$ and level 2 tropospheric CO from Sentinel-5P/Tropospheric Monitoring Instrument were also used to study the lockdown induced changes over Indian region. To investigate aerosol loading over Indian region, the daily gridded global Aerosol Optical Depth product (level 3) from the MODIS sensor on-board Terra (MOD08_D3_v6.1) and Aqua (MYD08_D3_v6.1) platforms were used. We have considered the quality flags as per the Algorithm Theoretical Basis Document for the respective sensors. As suggested python code is provided in appendix. Regarding the data, all the data resources are from the respective sites which provided in the Table 1. As suggested a few of the processed files are uploaded in the online repository (https://github.com/aarathimuppalla/airpollution_ld_study.git).

The Python code will explain the details of the processing method. It contains the metadata information namely version of the data used and their respective quality flags.

References for ATBD's

1. OMI Algorithm Theoretical Basis Document Volume IV OMI Trace Gas Algorithms, ATBD-OMI-02, Version 2.0, August 2002. https://eospso.gsfc.nasa.gov/atbd-category/49.

2. TROPOMI Algorithm Theoretical Basis Document of the total and tropospheric NO$_2$ data products, document no: S5P-KNMI-L2-0005-RP, http://www.tropomi.eu/documents/atbd, document no : S5P-KNMI-L2-0005-RP.
3. Algorithm Theoretical Baseline Document for Sentinel-5 Precursor: Carbon Monoxide Total Column Retrieval, http://www.tropomi.eu/documents/atbd, document number : SRON-S5P-LEV2-RP-002.
4. MODIS Atmosphere L3 Gridded Product Algorithm Theoretical Basis Document (ATBD) and users guide, Document No. ATBD-MOD-30 for Level-3 Global Gridded Atmosphere Products and Users Guide (Collection 6.1, Version 4.5, 06 August 2020. (https://atmosphere- imager.gsfc.nasa.gov/products/daily/documentation). The same has been updated in Section 2, Data in the revised manuscript.

Numbers in Figure 10 are not easy to read. Just add a table either in the manuscript or in a Material containing percentage changes in TCN and AOD for individual states for different phases.

Reply: Thank you for the suggestion. As suggested, Figure 10 is replaced with Table 3 in the revised manuscript.

Minor comments:

1. Please correct all the grammatical mistakes.

Reply: All the grammatical mistakes were corrected in the revised manuscript.

2. Affiliation looks very odd, there is comma after the name. Is it necessary to mention so many Divisions? Have a look at some good reference paper and follow the standard.

Reply: Thank you for the suggestion. The same has been corrected in the revised manuscript.

3. There is only one TROPOMI instrument so just say define S5P-TROPOMI once and the use just TROPOMI throughout, same with OMI. For MODIS, you can try to keep corresponding platform names.

Reply: Thank you for the suggestion. The same has been implemented in the revised manuscript.

4. Line 769: Try to use some standard tool to manage the references. Alono et al., should be placed somewhere at the top.

Reply: Thank you for the suggestion. The references are now kept in proper manner as per the journal format in the revised manuscript.

5. Line 16: May2020 – May 2020

Reply: The same has been corrected in the revised manuscript.

6. Line 17: Phase IV

Reply: The same has been corrected in the revised manuscript.

7. Line 19 : sectors "space" were halted during lockdown (LD) "space" which

Reply: The same has been corrected in the revised manuscript.

8. Line 20 : followed then "space" (phase-I and phase-II)

 Reply: The same has been corrected in the revised manuscript.

9. Lines 34-25 –Aqua and Terra are platforms not satellite instruments.

Reply: The same has been corrected in the revised manuscript.

10. Line 45: Tian et al., 2020 is about China, so say (e.g. Tian et al., 2020)

Reply: The same has been corrected in the revised manuscript.

11. Line 54: burnings

Reply: The same has been corrected in the revised manuscript.

12. Line 72 : remove "only". Also use appropriate reference for both as CO emission sources and lifetime estimates that are known for few decades.

Reply: The same has been incorporated in the revised manuscript.

13. Aerosols in not correct. Use "aerosol" throughout the manuscript.

Reply: The term aerosol has been used throughout in the revised manuscript.

14. Line 88 to 90- Are you discussing about India?

Reply: No. It is about the lockdown induced changes across the globe. For example, at China, Xu et al. (2020) reported $NO_2$ concentrations were 36.5% lower during lockdown period in comparison to the same period in previous year. In United States during lockdown period 25.5% decline in surface $NO_2$ concentrations was reported (Berman and Ebisu (2020). Spain showed reductions of 62% in $NO_2$ concentrations (Baldasano, 2020). Brazil also reported decline in $NO_2$ concentrations by 34-68% (Krecl et al., 2020). In India also studies reported reduction in $NO_2$ emission which is due to restricted human activities during lockdown period (Biswal et al., 2021).

15. Line 93: Follow ACP guideline for citation.

Reply: ACP guideline is followed for citation in the revised manuscript.

16. Line 100-103: Confusing sentence, reword it.

Reply: The sentence has been modified in the revised manuscript.

17. Line 110: Just sharpen the sentence to say what is new in this study.

Reply: The sentence has been revised as suggested.

18. Line 126: Write bit better description. Also write Version numbers and proper references (e.g. Lamsal et al., 2021, AMT for OMI V4 data or if you are using it.)

Reply: Thank you for the suggestion. Version number and Data levels are updated in the revised manuscript at Table 1.

19. Line 143: As per table 1, it should be ERA5. ERA-interim stops in 2019.

Reply: Thank you for the suggestion. The same has been modified in the revised manuscript.

20. Line 147: January 2014 should be enough if you don't want to write 31$^{st}$

Reply: Thank you for the suggestion. Period has been written as January 2014 to October 2020. The same has been updated in the revised manuscript.

21. Line 149: You do not detrend data to remove interannual variability. You detrend it to remove long-term changes.

Reply: We agree with your suggestion. We de-trend data to remove long- term changes. The same has been corrected in the revised manuscript.

22. Line 153: Do you mean individual states in India?

Reply: Yes, as shown in Table 3 in the revised manuscript.

23. Line 155: What is climatological. Monthly/weekly or daily mean values for all years are subtracted from the daily values? OK, I noticed on line 220 that you are using daily value. But processing description is somewhat scattered (sometimes it is daily and sometimes it is weekly or averaged over different period. Try to be bit more consistent.

Reply: Thank you for the comment. Daily values are used for computing the short-term climatological means during the lock period. Subsequently data are being used in different ways in order to understand the lockdown changes.

Line 155 : were calculated

Reply: The same has been corrected in the revised manuscript.

24. Line 159: remove "which was"

Reply: The same has been corrected in the revised manuscript.

25. Line 162 : Do you mean quality flags?

Reply: The line 162 refers to quality flags in the revised manuscript instead of quality factors.

26. Line 163: reword it. Region of interest used twice in once sentence. Do you mean your resample quality flags as well?

Reply: Thank you for the suggestion. The sentence has been corrected in the revised manuscript as swath and mask are calculated for the region of Interest and the data is resampled using nearest neighbourhood algorithm.

27. Line 168: thereafter or Then after?

Reply: Thereafter.

Line 219: Again, detrending does not remove inter-annual variability.

Reply: The same has been corrected to long - term changes as suggested in the revised manuscript.

28. Line 228: if you Redefining TCN and do it for TCC or don't do it altogether.

Reply: Thank you for the suggestion. TCN and TCC have been used throughout in the revised manuscript.

Line 236 – Figure 2: use slash (/) for dates at figure titles. It is somewhat confusing to read 01.01.20-07.01.20 that 01/01/20 to 07/01/20.

Reply: Thank you for the suggestion. The same has been corrected in all the figures.

29. Line 240-241- Seems to be far fetched. Do you have any evidence?

Reply: Thank you for the comment. During pre-LD time of 2020 year there was a small amount of reduction in TCN over the hotspot regions. However this reduction could be due to the inter-annual variability and seasonal component of $NO_2$ (Biswal et al., 2021). Also, across the globe, major cities have recorded reduction of $NO_2$ during the respective lockdown periods, which was just before the India's lockdown period (Tian et al., 2020; Xu et al., 2020; Berman and Ebisu, 2020; Collivignarelli et al., 2020; Jephcote et al., 2021). Hence a small reduction of TCN was observed during pre-LD.

30. Line 259: Are you sure of steel and cement industries near NCR . If it is true, it would be great if you can give some web link with the industry name and their capacities. I tried to find some information in the reference that was included but no luck.

Reply: Thank you for the suggestion. The author Garg et al. (2002) in his study on large point source emissions (thermal power plants, steel plants, cement plants, industrial processes such as sugar and paper and fossil fuel extraction) shares the details about the steel and cement industries distributed across India.

Line 260: What about central west region (e.g., Gujarat), it shows significant increase during phase I LD.

Reply: Thanks for the observation. Yes, an increase of TCN observed in the few part of central western region. This increase could be due to continuous operation of thermal power plants in the industrial corridor of Gujarat and presence of petroleum refineries. Same has been updated in the revised manuscript.

31. Line 263 – were given

Reply: The same has been corrected in the revised manuscript.

32. Line 275: sea breeze effects are ok for coastal area but inlands (e.g. Hyderabad, Bangalore)? Also confusing explanation. Why this effect should be minimum during normal years? I think it must be meteorology in the South India is way different than the North India or it might be due to sampling error.

Reply: We agree with your suggestion. $NO_2$ column amounts are lower over South India when compared to North India due to following reason listed below: Southern part of India is comparatively hot and humid which will lead to high OH (hydroxyl) radical concentrations than Northern part of India. As a result, photolysis of $NO_2$ will increase which results in low $NO_2$ concentrations. Further, In South India, the number of large point sources, amount of biomass burning, and vehicular population are less compared to Northern part of India.

We have now revised the statement and updated in the revised manuscript.

33. Line 306- years – year

Reply: The same has been corrected in the revised manuscript.

34. Line 320 – authors -> Lal et al.,( 2020)

Reply: The same has been corrected in the revised manuscript.

---

## Author Response (AR2)

Comments to the Author:

Dear authors,

based on the recommendation by the referees, I am happy to accept your paper for publications.

However, I would strongly recommend that you follow the advice of the last referee comment and improve the wording and the grammar of the paper.

Congratulations

Rolf

**Reply:** Dear Editor, thanks for the acceptance. We now improved the wording and grammar of the revised manuscript.  Github link is also made available to the users.

**Referee comments:**

1. Github directory is empty.

**Reply:** Thanks for the comment. Github link is made available to the users

2. Still many grammatical mistakes. Please take help from some senior author, otherwise proof-reading team might change the sentences some strange ways.

**Reply:** We now corrected English and grammar of the revised manuscript.